# *“I Actually Don’t Know What HIV Is”*: A Mixed Methods Analysis of College Students’ HIV Literacy

**DOI:** 10.3390/diseases8010001

**Published:** 2020-01-02

**Authors:** Robert M. Avina, Mathew Mullen, Salome Mshigeni, Monideepa B. Becerra

**Affiliations:** 1Department of Health Science and Human Ecology, California State University, San Bernardino 5500 University Parkway, San Bernardino, CA 92407, USA; robert.avina@csusb.edu (R.M.A.); 005231716@coyote.csusb.edu (M.M.); 2Department of Health Science and Human Ecology, Center for Health Equity, California State University, San Bernardino 5500 University Parkway, San Bernardino, CA 92407, USA; salome.mshigeni@csusb.edu

**Keywords:** HIV, literacy, prevention, STIs, college students, self-efficacy, knowledge, attitude

## Abstract

Objective: Human immunodeficiency virus (HIV) remains a major public health issue with young adults facing a disproportionately higher rate of the burden. Our goal was to address the current literacy related to HIV, including biomedical prevention methods and barriers to care, such as cultural factors, from a sample of college students. Methods: We conducted a convergent parallel mixed methods analysis where both qualitative and quantitative data were collected and analyzed separately. A thematic analysis was conducted to assess qualitative results, while descriptive statistics were conducted to assess quantitative survey results. Results: HIV literacy was limited, with several participants reporting that they did not understand what HIV meant. While the majority knew the use of condoms, knowledge of other biomedical prevention methods was limited, as was the understanding the risk factors of HIV, with participants noting sharing a toilet seat and promiscuity as risk factors. Self-efficacy for HIV prevention was low among participants with many reporting that cultural barriers prevented discussion of risky behaviors and prevention methods in their families and social groups. Conclusion: Targeted, culturally sensitive, health education initiatives are needed to understand the high, low, or no risks of HIV as well as address stigmas related to HIV.

## 1. Introduction

The human immunodeficiency virus (HIV) continues to be of public health importance in the United States. According to the most recent data from the Centers for Disease Control and Prevention (CDC), 38,739 people were diagnosed with HIV in 2017 [1]. Furthermore, HIV Surveillance Report highlight that young adults, ages 20–29 years, have the highest rate of such diagnoses in the United States, and this population is also more likely to be unaware of their status [2]. Likewise, in 2016, 8451 young adults in the United States were diagnosed with HIV, in which majority of those cases were among those aged 20–24 years and racial/ethnic minorities. For example, 54% were African Americans, 25% were Hispanics/Latinos, 16% were Whites, and 5% accounted for other race/ethnicities [3]. Given the disproportionate burden of such a preventable outcome shared among young adults, it is critical to understand the determinants.

The CDC notes that a plethora of factors, such as inadequate sex education, lack of condom use, having multiple sexual partners, etc., may contribute to this burden [3]. The literature also notes that many college students born after the AIDS crisis have a lack of knowledge and understanding regarding HIV, further leaving a large portion of the population susceptible to HIV transmission [4]. Recent findings from the CDC further suggested that young people were most likely to be unaware of their infection status (51%) and this is assumed to be due to their lack of the general understanding of what HIV is, and how it is transmitted [2].

Undoubtedly, health literacy is critical to healthy outcomes. Latest health literacy assessment, however, highlight that only 12% of the United States’ adult population exemplify proficient health literacy, indicating that 77 million people in the nation would have difficulty with common health tasks [5]. Despite such a critical social barrier, health literacy assessment in the nation is dated and little evidence exists on HIV literacy, especially one based on understanding common myths, perceived severity and susceptibility, as well as self-efficacy and cultural barriers to care. As such, the goal of this paper was to address the current HIV literacy at a mid-sized college campus, with a primarily first-generation minority student population. 

## 2. Methods

### 2.1. Study Design

We employed a convergent parallel mixed methods approach. As such, quantitative and qualitative data were collected and analyzed separately and results were compared and contrasted [6]. Specifically, the quantitative data were collected in two rounds to ensure adequate sample size and a total of three focus groups were conducted to obtain more in-depth content. 

### 2.2. Sample and Data Collection

Our study focused on college students at a mid-sized minority-serving institution. The current student population is predominantly from the Inland Southern California region, with the majority being females (61%), first generation college students (81%), and Hispanics (63%) [7]. Participants were recruited from general education courses in order to obtain diversity and representation of the university. Extra credit points were provided as an incentive upon approval of the Institutional Review Board. All students were eligible to participate in the study as long as they were currently enrolled at the institution. The University’s Institutional Review Board Approved two rounds of quantitative data collection of a maximum of 500 students per round. This quantitative phase consisted of two rounds of data collection through survey questionnaires. Recruitment of participants for focus groups continued until theoretical saturation was reached. 

For quantitative phase, the second round of questions was developed based on trends noted in round 1. Such questions were divided into the following broad categories: perceived severity, perceived susceptibility, and self-efficacy of HIV prevention, in addition to basic HIV prevention knowledge. We chose not to assess HIV prevention practices as the current study focused on understanding HIV literacy with the aim to follow up on assessing practices and implementation of appropriate health education strategies. The central questions for the qualitative phase were primarily focused on understanding HIV prevention methods, including biomedical prevention options. Follow up probe questions were based on answers given by participants. 

### 2.3. Data Analysis

For the quantitative phase, we used descriptive statistics to evaluate the prevalence of correct responses for each question related to the key constructs of interest. All statistical analyses were conducted in SPSS version 24 (IBM, Corp., Armonk, NY, USA). Qualitative responses were digitally recorded, transcribed verbatim, and thematically analyzed, where the codes and thematic schemes were evaluated in the context of knowledge, perceived severity, susceptibility, and self-efficacy of HIV prevention. There are no public use data for this study.

### 2.4. Results

A total of 428 students were included for quantitative assessment and five students were included for focus groups as theoretical saturation was reached after the fifth participant. 

## 3. HIV General and Prevention Knowledge 

Quantitative assessment of HIV knowledge (Table 1) demonstrates that 35.6% of the participants did not know what HIV meant and only 30.3% of the participants correctly identified how it impacts the immune system. Qualitative assessment further confirmed such limited HIV knowledge, for example, some of the feedback stated,
“I actually don’t know what HIV is.” 
“I don’t know exactly what it is, but I know that it’s a transmitted disease that you get.”

HIV prevention knowledge was also limited. While a majority (75.5%) of the participants correctly identified that a combination of latex condom and PrEP was most effective in HIV prevention, another 22.2% still reported “*don’t know*.” Knowledge on PrEP in general was also low, with the majority of the participants unable to correctly identify the use of PrEP or the population that this method is primarily targeted for. For example, 68% of the participants said “*don’t know*” to whether PrEP can prevent HIV if taken daily. Likewise, 67.3% reported “*don’t know*” when asked if PrEP is only for people who are known to be HIV negative. HIV prevention through means of condom was also varied with only 61.3% correctly noting that latex condoms were more effective than natural skin in preventing HIV. 

Likewise, qualitative assessment showed limited general understanding of HIV prevention. For example, when asked to provide ways to prevent HIV, participants stated: 

“Just use condoms and get tested frequently and be clean.”

“[I know] very little [about HIV prevention], I know about regular things like contraceptives. Like using condoms to prevent getting HIV.” 

“Contraceptives, right? Is that what condoms are called?”

### 3.1. Attitude and Social Norms 

When assessing attitude (Table 2) towards HIV compared to other STIs, 52.6% either reported “*don’t know*” or agreed with the myth-based statement that those who get HIV are more promiscuous than those who get other STIs. This was further seen in qualitative assessment with statements such as:

“Having an immense amount of sex partners and not cleaning yourself after”.

“…not cleaning the condom before taking if off.”

“One-night stands.”

“When you just meet someone and just end up sleeping with them.”

“..it also speaks negatively about the person that they are irresponsible or that they sleep around.”

When asked if it is easier to get HIV than other STIs, again, 64.7% either reported “*don’t know*” or said “*true.*” While one participant did bring up HIV being more prevalent among gay communities, it was not a predominant theme in the focus groups or quantitative survey.

A consistent theme that came up in the focus groups was that of social norm when discussing attitude towards HIV, especially related to culture. Throughout the focus groups, participants noted that their social circle, often heavily influenced by their cultural norms, viewed HIV as a topic that has a negative aspect of sexual practice and thus was not discussed. 

“...as far as my social cultural goes I think that HIV is perceived as not only this very negative thing.” 

“I just think they see it as something negative it’s not something to be happy about and they look down on it.” 

“It’s a very victim type blaming thing so it like people who have HIV brought upon themselves because they were irresponsible in their sexual habits.” 

### 3.2. Perceived Severity

In addition to HIV knowledge, perceived severity (Table 3) of HIV was low among the population, though we found basic biological severity knowledge was understood. For example, results of quantitative assessment demonstrate that the majority of the participants correctly identified that there was no cure for HIV (90.1%) and the biological mechanism of AIDS (89%). Likewise, while a majority correctly noted that HIV was the virus causing AIDS (89%), an emergent subtheme in focus groups was the lack of understanding how severe HIV can become.

“I really don’t know what the actually difference is all I know is that HIV leads to AIDS.” 

“Like once you have HIV you’re going to get AIDS and you’re going to die.”

“It means that you can take antibiotics and it will go away.”

“Like I don’t know the HIV symptoms but if I notice a bump down there or like if you see certain things that don’t belong there it’s like ok somethings wrong with me let me go get tested.”

### 3.3. Perceived Susceptibility 

Both qualitative and quantitative data demonstrate a higher perceived susceptibility of HIV (Table 4) in the population, though understanding of risk factors continued to remain low. For example, nearly 86% of the population noted that there was no vaccine to prevent HIV. In qualitative assessment, participants further stated that they were often worried about not knowing their HIV status.

“You just have to keep worrying about it.” 

“I think that it just effects your daily routines.” 

Risk factor identification, however, was varied in the population. For instance, the majority of the study participants identified sharing needles, various forms of sex (anal or oral), and maternal to child transfer of HIV. 

On the other hand, only 51.6% of the study population stated false to the myth-based statement that “*a person can get HIV from a toilet seat.*” Likewise, only 60.1% of the participants correctly identified HIV mode of transmission. While the majority did not believe kissing could spread HIV, given that nearly 25% did stated it was true, we wanted to further add whether touching, in addition to kissing, was considered a risk factor in the second round of data collection, and 36.6% reported either “*don’t know*” or “*true.*”

This was further confirmed in qualitative assessment, where “*unprotected* sex” was often identified as a risk factor, participants also noted that they did not know much beyond it or often referred to myths about HIV risk. 

“[HIV] can be transmitted sexually or by cuts or anything like that.”

“So, if you get a cut then and anyone who has HIV get a cut then can I get it?” 

“Like with anti-bacterial soap or stuff like that or not cleaning the condom before.” 

### 3.4. Self-Efficacy

As shown in Table 5, participants were asked about their confidence in their knowledge of HIV. A 43.4% reported *strongly agree/agree*. Another 58.5% and 80.2% also reported *strongly agree/agree* to their confidence in being able to identify risk factors and prevent HIV, respectively. On the contrary, majority reported *strongly disagree/disagree* in their confidence in PrEP knowledge. 

## 4. Discussion

This study aimed to assess HIV literacy among college students, with an emphasis on preventive measures and risk factors, as prior research indicates that many college students lack a general understanding of the disease [4]. Our study has several key findings that warrant further discussion: (1) a large portion of study participants were not aware of what HIV meant, (2) myths about HIV transmission were common, (3) limited knowledge of biomedical prevention methods was prevalent, (4) self-efficacy for HIV prevention was low, and finally, (5) cultural norms served as barriers to HIV literacy.

Our results show that over 30% of the population did not know what HIV stood for, and a majority were not aware of its impact on the body immune system. A recent study by Andrew et al. [8] conducted among students at a historically black campus in the United States, noted that nearly 97% of the study population had good knowledge of HIV/AIDS. Similar trends of high HIV/AIDS knowledge was found among another historically black campus [9]. On the other hand, another study among Chinese students in the United States noted misinformation about HIV/AIDS [10], which was similar to our findings. Such contradictory findings in the literature could be attributable to cultural differences of recent immigrant populations. In our study, a majority of the population was Hispanic. Similar to the study focused on Chinese students, Hispanic families, as noted in our study results, have cultural stigma against discussion of sexual health. Such stigma, in turn, can lead to misinformation of HIV/AIDS. For example, a study addressing HIV care among Hispanics [11] noted that misconceptions and stigma related to HIV/AIDS was prevalent, especially among those who were less acculturated.

Similar to Andrew et al. [8], we also noted that students in our population were not well aware of modes of HIV transmission. Our study also looked more in depth to evaluate the most common myths of HIV transmission, including sharing toilet seat, kissing, touching, etc. Such results highlight that comprehensive HIV prevention specific educational initiatives are needed, especially one that discusses high-low-no modes of HIV transmission. Likewise, while knowledge of condom use as a means to prevent HIV was prevalent, little literacy existed on biomedical prevention methods. In a study among gay and bisexual men, Liu et al. [12] noted that only 16% of the study participants reported awareness about PrEP. Despite the scientific growth and marketing of such prevention methods, our study also corroborates the low knowledge of such resources. In a study to assess willingness to use PrEP among African-American college students, the authors found that over 80% of study participants were willing to take PrEP if they knew the medication would lower risk of HIV, knew that they themselves were HIV positive, or their partner was HIV positive [13]. While similar studies among other racial/ethnic groups, especially those with cultural barriers do not exist in the United States, college health educators can create opportunities for open dialog to ensure that young adults are knowledgeable regarding all available HIV prevention methods.

Additionally, Burns and Dillon [14] noted that among college students, self-efficacy was associated with higher probability of condom use. In our study, however, we found that majority of the population reported low self-efficacy to HIV prevention. Given the importance of such a construct in adequate preventive measures, health education initiatives that improve college students’ self-efficacy in condom use, as well as biomedical prevention methods, are critical.

Cumulatively, the literature, in addition to our study results, highlights that college students continue to have low levels of literacy on key aspects of HIV, especially prevention and transmission methods. Based on such evidence, we postulate that low HIV literacy is more prevalent when cultural barriers interplay with sexual health discussions, which in turn can lead to lower self-efficacy. Nearly two decades after the discovery of Highly Active Antiretroviral Therapy (HAART) and six years after the Federal Drug Administration (FDA) approval of PrEP for prevention purposes, disparities still exists, not only in terms of age and ethnicity, but also on the literacy component, and college students are not an exception [15]. A study conducted to assess HIV knowledge among a specific group of low-income African-American adolescents found out that having accurate information about HIV may benefit sexual health outcomes and consequently encourage positive attitudes towards safe sex practices [16], in turn demonstrating the importance of HIV education. Evidently, improving HIV literacy among college students can further alleviate the disproportionate burden of the disease shared among young adults.

Thus, this study brings to light a potential forgotten public health issue among college students that demonstrates how health literacy, especially one related to HIV, remains to be a crucial need that improves overall health outcomes. The first of December every year is World AIDS Day, the only day that people around the world unite in the fight against HIV/AIDS, to show their support for not only those who have died, but also for those who are living with HIV [17]. Hence, there is ample opportunities for colleges to effectively use this day and raise awareness of HIV/AIDS in order to alleviate the HIV burden among young adults.

The results of this study should be interpreted in the context of its limitations. Self-reported data are subject to recall bias. Given the sensitive nature of this topic, some participants may not have felt comfortable answering some questions and there could be potential underreporting. Likewise, we did not assess whether acculturation, parental education, or having a family member with HIV impacted HIV literacy and are thus scopes for further research.

Notwithstanding such limitations, the results of our study highlight that HIV prevention must be a major public health priority, especially among young adults who may have cultural and social norms that promote myths about HIV transmission and risk. In addition, while our study highlights that college students are aware of the basics of HIV prevention, such as condom use, biomedical intervention knowledge is limited, as is differentiating between the high, no, and low risk of HIV transmission. This result is critical as myths regarding promiscuity continue to be prevalent in the study population, as are misconceptions regarding how HIV spreads. It is imperative that to truly reduce to zero rate for HIV [18], such myths, which may stem from lack of family discussion on sexual health, must be targeted during general health education and sexual health education courses. Targeted HIV health education initiatives that address cultural and social norms that may perpetuate myths are of critical need.

## Figures and Tables

**Table 1 diseases-08-00001-t001:** HIV knowledge among college students (n = 428).

Question	Answer Options	Round 1 Percent (%)	Round 2 Percent (%)
Knowledge on HIV and prevention			
What does the acronym HIV stand for?	Don’t Know		15.0%
	Human Immobilization Virus		17.0%
	Human Immunodeficiency Virus		63.4%
	Hemoglobin Insufficiency Virus		4.6%
What does HIV do to the body once it is in the blood stream?	Don’t Know		19.1%
	Attacks T-helper cells		30.3%
	Attacks red blood cells		48.7%
	Attacks non-ciliated epithelial cells		2.0%
HIV is the virus that causes AIDS.	True	89.0%	
	False	11.0%	
Taking a test for HIV one week after having sex will tell a person if she or he has HIV.	Missing	0.4%	
	True	36.5%	
	False	63.1%	
Which of the following is a preventative measure from HIV transmission?	Don’t know		16.4%
	Using latex condoms in combination with PREP when engaging in sexual intercourse		75.7%
	Using lambskin condoms when engaging in sexual intercourse		4.6%
	Using only birth control when engaging in sexual intercourse		3.3%
	Don’t know		22.2%
Latex condoms are more effective than natural skin condoms in preventing HIV.	Missing	0.4%	
	True	61.3%	
	False	38.3%	
PReP can prevent HIV if taken daily.	Don’t know		68.0%
	True		22.2%
	False		9.8%
PReP can cure HIV if taken daily.	Don’t know		52.9%
	True		2.6%
	False		44.4%
A HIV positive person can be on PReP.	Don’t know		70.6%
	True		19.6%
	False		9.8%
PReP is only for people who are known to be HIV negative.	Don’t know		67.3%
	True		11.1%
	**False**		**21.6%**

**Table 2 diseases-08-00001-t002:** HIV-related attitude among college students (n = 428).

Question	Answer Options	Round 1 Percent (%)	Round 2 Percent (%)
Attitude			
People who get HIV are more promiscuous than those who get other STIs.	Don’t know		32.2%
	True		20.4%
	False		47.4%
It is easier to get HIV than other STIs.	Don’t know		50.3%
	True		14.4%
	False		35.3%

**Table 3 diseases-08-00001-t003:** Perceived severity of HIV among college students (n = 428).

Question	Answer Options	Round 1 Percent (%)	Round 2 Percent (%)
Perceived Severity			
There is a cure for AIDS.	Missing	0.4%	
	True	9.6%	
	False	90.1%	
AIDS is a medical condition in which your body cannot fight off diseases.	True	89.0%	
	False	11.0%	
A person who is HIV-positive has AIDS.	True	38.7%	
	False	61.3%	
People who have been infected with HIV quickly show serious signs of being infected.	True	11.7%	
	False	88.3%	

**Table 4 diseases-08-00001-t004:** Perceived severity of HIV among college students (n = 428).

Question	Answer Options	Round 1 Percent (%)	Round 2 Percent (%)
Perceived Susceptibility			
There is a vaccine that can stop adults from getting HIV.	True	14.5%	
	False	85.5%	
A person can get HIV from a toilet seat.	Don’t know		22.9%
	True		25.5%
	False		51.6%
A person can get HIV from sharing needles.	Don’t know		9.2%
	True		88.8%
	False		2.0%
Kissing can spread HIV.	True	24.5%	
	False	75.5%	
A person can get HIV from touching and kissing.	Don’t know		15.0%
	True		21.6%
	False		63.4%
HIV can only be transmitted sexually.	Don’t know		10.5%
	True		9.2%
	False		80.4%
Which of the following is considered modes of transmission for HIV?	Don’t know		7.8%
	Blood, semen/precum, breastmilk, saliva		25.5%
	semen/precum, nasal secretions, breast milk, blood		6.5%
	vaginal secretions, blood, semen/precum, breast milk		60.1%
A woman can pass HIV to her baby during childbirth	Don’t know		12.4%
	True		75.8%
	False		11.8%
A woman cannot get HIV if she has sex during her period	True	4.3%	
	False	95.7%	
Having oral sex instead of sexual intercourse prevents HIV	True	5.0%	
	False	95.0%	
Having anal sex instead of sexual intercourse prevents HIV	True	3.9%	
	False	96.1%	
A person with HIV can spread it to others even before they get AIDS	True	88.3%	
	False	11.7%	
Having another sexually transmitted infection can increase a person risk of getting HIV	True	71.6%	
	False	28.4%	
Using alcohol or drugs before or during sex can increase a person’s risk of getting HIV	True	40.4%	
	False	59.6%	
A person can get HIV by sitting in a hot tub or a swimming pool with a person who has HIV	True	13.8%	
	False	86.2%	
A mother who is HIV-positive can infect her child through breast milk	Missing	0.4%	
	True	77.0%	
	False	22.7%	
All pregnant women infected with HIV will have babies born with AIDS	True	23.0%	
	False	77.0%	

**Table 5 diseases-08-00001-t005:** HIV self-efficacy among college students (n = 428).

Question	Answer Options	Round 1 Percent (%)	Round 2 Percent (%)
Self-Efficacy			
I am confident in my knowledge of HIV	Strongly agree		6.6%
	Agree		36.8%
	Disagree		40.1%
	Strongly disagree		16.4%
I am confident in my knowledge of PrEP	Strongly agree		2.0%
	Agree		7.9%
	Disagree		41.4%
	Strongly disagree		48.7%
I am confident in my ability to prevent HIV infection	Strongly agree		38.8%
	Agree		41.4%
	Disagree		14.5%
	Strongly disagree		5.3%
I am confident in my ability to distinguish between the high, low, and no’s of HIV risk	Strongly agree		21.7%
	Agree		36.8%
	Disagree		25.7%
	Strongly disagree		15.8%

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
