# Peer review of "“I Actually Don’t Know What HIV Is”: A Mixed Methods Analysis of College Students’ HIV Literacy"

_diseases, 2020, doi:10.3390/diseases8010001_

Round 1
Reviewer 1 Report
This is an interesting paper, by Avina and colleagues, assessing the HIV literacy among college students. The subject is not novel (Diseases 2018, 6(4), 98; https://doi.org/10.3390/diseases6040098), however the data obtained reinforces previous studies describing the lack of knowledge in HIV disease. The paper is well written with interesting points. But, a better description of the students should be included. In line 209, the period should be removed after "associated with".
Author Response
Reviewer 1 feedback:
Comments and Suggestions for Authors
This is an interesting paper, by Avina and colleagues, assessing the HIV literacy among college students. The subject is not novel (Diseases 2018, 6(4), 98; https://doi.org/10.3390/diseases6040098), however the data obtained reinforces previous studies describing the lack of knowledge in HIV disease. The paper is well written with interesting points. But, a better description of the students should be included. In line 209, the period should be removed after "associated with".
Response: We have added a summary of the student population in the methods section from lines 92-94. We have also removed the period after “associated with,” which is now line 244
Reviewer 2 Report
Dear Authors,
I appreciate the authors for their manuscript, however certain things need to be addressed as follows;
would you add assessment knowledge of the participants related to 1. preventive measures, 1. mode of infection routes, 3. first-aid therapy? whats outcome of this knowledge-based study ? would you add significance of this manuscript in the description in more detailed manner?Thanks,
LR
Author Response
Reviewer 2 feedback:
Dear Authors,
I appreciate the authors for their manuscript, however certain things need to be addressed as follows;
would you add assessment knowledge of the participants related to 1. preventive measures, 1. mode of infection routes, 3. first-aid therapy? whats outcome of this knowledge-based study ? would you add significance of this manuscript in the description in more detailed manner?
Thanks,
LR
Response: In our results and discussion, we have detailed assessment of knowledge among the participants, in relation to preventive measures (lines 127-141) and mode of infection routes (line 187-202). However, we did not address first-aid therapy as that was not a foundation of our study. We also added a section on outcome of this knowledge-based study in the discussion, along with more detailed significance of the manuscript (lines 261-267 and 278-283).
Reviewer 3 Report
1) This paper reports on a well-designed and conducted mixed method surveillance study to determine college students' knowledge of the risk of HIV infection from sexual and drug injection exposure practices in the campus environment
2) To improve understanding of HIV infection and HIV disease risk in the University community, the investigators conducted a simultaneous convergence parallel mixed method analysis of the qualitative and quantitative data on the students' knowledge of HIV infection and disease risk. From the analysis they determined that: 1) Knowledge of HIV infection and disease was limited in the student population, 2) Cultural barriers limited the discussion of risk behaviors and the design and conduct of prevention programs, and 3) Targeted culturally sensitive health education initiatives are needed to improve understanding of the risk of HIV infection among college student populations.
3) This manuscript evaluates: a) college students' knowlege of sexually and IV drug use transmitted HIV infection, and b) the associated infection and disease risks and the methods and the practices that can protect college students against becoming infected with the virus. Such epidemiological data are critical for designing effective HIV prevention programs for HIV infection and the development of AIDS in this vulnerable age group.
4) The manuscript title, which starts with a quote from a study participant's response to one of the study questions, denotes a low state of knowledge and the lack of understanding of the risks of HIV infection and disease among college-age students in the United States.
6) To improve understanding of the HIV infection risk and HIV disease, the investigators conducted a simultaneous convergent parallel mixed method analysis of qualitative and quantitive data on the students' knowledge of HIV infection and disease risk. From the data, they concluded that: 1) Knowledge of HIV infection risk and disease was limited in the student population and that cultural barriers limited discussion of risk behaviors and the design and conduct of prevention programs, and 2) targeted culturally sensitive health education initiatives are needed to improve understanding of the risk of HIV infection among college student populations.
7) The manuscript needs editing for English grammar, especially for incomplete sentences and spelling
Author Response
Reviewer 3:
1) This paper reports on a well-designed and conducted mixed method surveillance study to determine college students' knowledge of the risk of HIV infection from sexual and drug injection exposure practices in the campus environment
2) To improve understanding of HIV infection and HIV disease risk in the University community, the investigators conducted a simultaneous convergence parallel mixed method analysis of the qualitative and quantitative data on the students' knowledge of HIV infection and disease risk. From the analysis they determined that: 1) Knowledge of HIV infection and disease was limited in the student population, 2) Cultural barriers limited the discussion of risk behaviors and the design and conduct of prevention programs, and 3) Targeted culturally sensitive health education initiatives are needed to improve understanding of the risk of HIV infection among college student populations.
3) This manuscript evaluates: a) college students' knowlege of sexually and IV drug use transmitted HIV infection, and b) the associated infection and disease risks and the methods and the practices that can protect college students against becoming infected with the virus. Such epidemiological data are critical for designing effective HIV prevention programs for HIV infection and the development of AIDS in this vulnerable age group.
4) The manuscript title, which starts with a quote from a study participant's response to one of the study questions, denotes a low state of knowledge and the lack of understanding of the risks of HIV infection and disease among college-age students in the United States.
6) To improve understanding of the HIV infection risk and HIV disease, the investigators conducted a simultaneous convergent parallel mixed method analysis of qualitative and quantitive data on the students' knowledge of HIV infection and disease risk. From the data, they concluded that: 1) Knowledge of HIV infection risk and disease was limited in the student population and that cultural barriers limited discussion of risk behaviors and the design and conduct of prevention programs, and 2) targeted culturally sensitive health education initiatives are needed to improve understanding of the risk of HIV infection among college student populations.
7) The manuscript needs editing for English grammar, especially for incomplete sentences and spelling
Response: We have thoroughly reviewed the manuscript and addressed the noted grammatical and spelling errors.